# Chemokine (C-C Motif) Ligand 8 and Tubulo-Interstitial Injury in Chronic Kidney Disease

**DOI:** 10.3390/cells11040658

**Published:** 2022-02-14

**Authors:** Jangwook Lee, Yeonhee Lee, Kyu-Hong Kim, Dong-Ki Kim, Kwon-Wook Joo, Sung-Joon Shin, Yon-Su Kim, Seung-Hee Yang

**Affiliations:** 1Department of Internal Medicine, Dongguk University Ilsan Hospital, Goyang-si 10326, Gyeonggi-do, Korea; dive2inf@gmail.com (J.L.); shine5050@naver.com (S.-J.S.); 2Department of Internal Medicine, Uijeongbu Eulji Medical Center, Eulji University, Uijeongbu-si 11749, Gyeonggi-do, Korea; 2yh870@gmail.com; 3Department of Internal Medicine, Seoul National University College of Medicine, Seoul 03080, Korea; dkkim73@gmail.com (D.-K.K.); junephro@gmail.com (K.-W.J.); yonsukim@snu.ac.kr (Y.-S.K.); 4Kidney Research Institute, Seoul National University, Seoul 03080, Korea; suda1106@naver.com; 5Biomedical Research Institute, Seoul National University Hospital, Seoul 03080, Korea

**Keywords:** CCL8, CCR2, fibrosis, apoptosis, chronic kidney disease

## Abstract

Kidney fibrosis has been accepted to be a common pathological outcome of chronic kidney disease (CKD). We aimed to examine serum levels and tissue expression of chemokine (C-C motif) ligand 8 (CCL8) in patients with CKD and to investigate their association with kidney fibrosis in CKD model. Serum levels and tissue expression of CCL8 significantly increased with advancing CKD stage, proteinuria level, and pathologic deterioration. In Western blot analysis of primary cultured human tubular epithelial cells after induction of fibrosis with rTGF-β, CCL8 was upregulated by rTGF-β treatment and the simultaneous treatment with anti-CCL8 mAb mitigated the rTGF-β-induced an increase in fibronectin and a decrease E-cadherin and BCL-2 protein levels. The antiapoptotic effect of the anti-CCL8 mAb was also demonstrated by Annexin V/propidium iodide staining assay. In qRT-PCR analysis, mRNA expression levels of the markers for fibrosis and apoptosis showed similar expression patterns to those observed by western blotting. The immunohistochemical analysis revealed CCL8 and fibrosis- and apoptosis-related markers significantly increased in the unilateral ureteral obstruction model, which agrees with our in vitro findings. In conclusion, CCL8 pathway is associated with increased risk of kidney fibrosis and that CCL8 blockade can ameliorate kidney fibrosis and apoptosis.

## 1. Introduction

Chronic kidney disease (CKD) is a major concern associated with high mortality and morbidity [1]. Once acute or chronic kidney injury exceeds a certain threshold, the progression of kidney disease is consistent, irreversible, and mostly independent of the initial insult [2]. The final common pathway in this process has been thoroughly investigated, and kidney fibrosis, which is characterized by a serial process of tubulointerstitial fibrosis, tubular atrophy, and glomerulosclerosis, has been accepted as a common pathological outcome of CKD, regardless of etiology [3,4,5,6,7,8,9,10]. During this process, hypoxia plays a pathogenic role in the relatively early stages of CKD, before the development of structural tubulointerstitial injury [2,11]. Subsequently, kidney ischemia is induced by the loss of peritubular capillaries in the tubulointerstitium during the late stage of kidney disease [11,12].

Transforming growth factor-β (TGF-β) has been reported as a key factor in regulating kidney fibrosis [9]. TGF-β is broadly expressed in almost every cell type and acts by regulating an intracellular signaling cascade involving the Smad family of proteins through the ligand-induced activation of heteromeric transmembrane TGF-β receptor kinases [7,13]. TGF-β can promote epithelial-to-mesenchymal transition (EMT) to become myofibroblasts of tubular epithelial cells (TECs) and alter the extracellular matrix metabolism of resident kidney cells [14,15,16].

However, reliable diagnostic biomarkers for kidney fibrosis are still lacking. Therefore, to identify new biomarkers of kidney fibrosis, we performed a prospective study of kidney tubule cell RNA sequencing from human with glomerulopathy. Among a large number of differentially expressed genes, chemokine (C-C motif) ligand 8 (CCL8) was significantly upregulated in most glomerulopathies with fibrosis, compared to minimal change disease and focal segmental glomerulosclerosis, which were fibrotic, and inflammatory processes were reduced in disease progression. This may be tangible evidence that chemokines and their receptors play a crucial role in the progression of CKD [17]. Therefore, we next investigated whether CCL8 could be used as a marker for clinical disease progression.

CCL8, also known as monocyte chemoattractant protein-2 (MCP-2), was first identified in human osteosarcoma cells and functions in a wide variety of inflammatory cells as a chemotactic factor by binding chemokine receptors including CCR2 [18,19,20]. Human CCL8 was only recently identified, and its elevation has been described in limited diseases, such as graft-versus-host disease (GVHD) [18], idiopathic pulmonary fibrosis (IPF) [21], and allergic dermatitis [22]. The serum CCL8 level in GVHD patients correlates closely with GVHD severity and is a promising specific serum marker for early and accurate diagnosis [18]. Moreover, a recent study suggested that CCL8 concentrations in bronchoalveolar lavage fluids and CCL protein expression in lung tissues were significantly higher in patients with IPF, and that high levels of CCL8 were associated with a reduced survival rate [21]. In 2013, Islam et al. also reported that CCL8, which is highly expressed in the skin, serves as an agonist for CC chemokine receptor 8 (CCR8) in mouse, and that the elevated CCL8 would be related with activation of CCR8 in chronic inflammation process in human [22].

However, none of the above studies have addressed the role of CCL8 in kidney disease. Furthermore, the serum or urine levels of CCL8 have not been explored in CKD patients. Therefore, we aimed to examine the serum levels and tissue expression of CCL8 in patients with CKD and to investigate their association with kidney fibrosis in CKD model.

## 2. Materials and Methods

### 2.1. Study Population and Data Collection

Patients with a diagnosis of IgAN, diabetic nephropathy (DN), and normal healthy controls, confirmed by kidney biopsy, were recruited at the Seoul National University Hospital between January 2015 and October 2020. Patients under 18 years of age, missing medical records, presence of active infection, and diagnosis of malignancies/rheumatologic disease were excluded. At the time of diagnosis, demographic and clinical features, including age, sex, body mass index (BMI); and biochemical parameters, such as serum creatinine (Cr), estimated glomerular filtration rate (eGFR), spot urine protein/creatinine ratio (uPCR), and other laboratory findings (serum hemoglobin, albumin, and high-sensitivity C-reactive protein levels), were collected from the patients’ electronic medical records.

### 2.2. Measurement of Serum CCL8 Levels by Enzyme-Linked Immunosorbent Assay (ELISA)

Serum levels of CCL8 (Lifespan Biosciences, Seattle, WA, USA) were quantified by using an enzyme-linked immunosorbent assay (ELISA) according to the manufacturer’s protocol. All measurements were performed in duplicate in a blinded manner. The lower limit of detection for CCL8 was 12.5 ng/mL.

### 2.3. Histopathologic Evaluations

Kidney tissue samples were evaluated by light, electron, and immunofluorescence microscopy and diagnosed by a nephropathologist. The degree of interstitial fibrosis and tubular atrophy was assessed. Interstitial fibrosis, tubular atrophy, and interstitial inflammation were scored based on the percentage of affected area as follows: 0, none; 1, mild, ≤25%; 2, moderate, 26–50%; and 3, marked, >50% or diffuse pattern.

### 2.4. Immunohistochemistry of Kidney Biopsy Samples

Eighty-three kidney tissue samples were processed for immunohistochemistry. Unstained tissue samples obtained from the study population were used. Paraffin-embedded kidney tissue samples were cut into 4-μm-thick sections, deparaffinized, and rehydrated using xylene and ethanol. After blocking the endogenous streptavidin activity using 3% hydrogen peroxide, the sections were stained with anti-CCL8 antibody (Ab) (Novus Biologics, Littleton, CO, USA) anti-CCR8 Ab (Invitrogen, Carlsbad, CA, USA) and anti-CCR2 Ab (Abcam, Cambridge, UK) and incubated at 4 °C overnight. Next, the samples were incubated with dextran polymer conjugated with horseradish peroxidase (GBI Labs, Bothel, WA, USA) for 5 min at room temperature. Finally, all sections were counterstained with Mayer’s hematoxylin (Sigma–Aldrich, St Louis, MO, USA) and examined by light microscopy (DFC-295; Leica, Mannheim, Germany). Quantification of CCL8 and CCR2-positive cells was performed using a computer-based morphometric analysis (Qwin 3 and LAS-4000, Leica, Mannheim, Germany). The intensity scoring was performed in a blinded manner by calculating the mean values of the positive areas (%). All analyses were reviewed and confirmed by the renal pathologist who had no knowledge of each experimental group.

### 2.5. Isolation and Primary Culture of Human Proximal Tubular Epithelial Cells

The protocol for obtaining and processing human kidney specimens was reviewed and approved by the Institutional Review Board of Seoul National University Hospital (IRB no. 1002-045-309). Human proximal tubule segments unaffected by tumors were isolated from the kidneys surgically removed from patients who were diagnosed with renal cell carcinoma. After dissecting the cortex, the minced specimens were digested with Hank’s balanced salt solution (HBSS) containing 3 mg/mL collagenase (Sigma-Aldrich) and incubated at 37 °C for 1 h. The digested kidney cells were washed through a series of sieves (120, 70, and 40 μm in diameter) using phosphate-buffered saline, followed by centrifugation at 500× *g* for 5 min. Human tubular epithelial cells (hTECs) were recovered from the pellet and incubated in DMEM/F12 for 4 h. Tubules floating in the media were gathered and cultured on collagen-coated Petri dishes (BD Biosciences, San Jose, CA, USA) until epithelial cell colonies were established. Cells were used after 2–3 passages. hTECs (2 × 10^5^ cells/well) were seeded in six-well plates. Following serum starvation, fibrosis was induced with 2 ng/mL of recombinant (r) TGF-β (R&D Systems, Wiesbaden, Germany) for 48 h. To evaluate the role of CCL8 in fibrosis, rTGF-β-stimulated hTECs were simultaneously treated with anti-CCL8 monoclonal antibody (mAb; Novus Biologics) at 1 μg/mL, or recombinant (r)CCL8 (R&D Systems) at 1 μg/mL. After 2 days of treatment, cell fibrosis was observed. In vitro experiments were performed at least twice.

### 2.6. Western Blot Analysis

After the hTECs were removed from culture dishes, the proteins were retrieved using radioimmunoprecipitation assay (RIPA) buffer containing Halt protease inhibitor (Pierce, Rockford, IL, USA). Western blotting was performed using primary antibodies against fibronectin (Abcam), CCL8 (Novus Biologics), E-cadherin (Abcam), BCL-2 (Santa Cruz Biotechnology, Dallas, TX, USA), CD44 (Abcam), CCR2 (Abcam), and β-actin (Sigma-Aldrich) CD44 (Abcam), a transmembrane receptor, interacts with cell-matrix components, such as fibronectin and collagen, and is a potential marker of fibrosis. Equal amounts of extracted proteins (20~40 μg) were isolated on 10% sodium dodecyl sulfate-polyacrylamide gels and transferred onto Immobilon-FL 0.4 μm of polyvinylidene difluoride membranes (Millipore, Burlington, MA, USA). Anti-rabbit IgG (Cell Signaling Technology, Danvers, MA, USA) and anti-mouse IgG (Cell Signaling Technology) were used as horseradish peroxidase-conjugated secondary antibodies. The immunoblot bands were visualized and images were captured using an ImageQuant LAS 4000 Mini instrument (GE Healthcare, Princeton, NJ, USA). Western blotting results were quantified using ImageJ (National Institutes of Health, Bethesda, MD, USA).

### 2.7. Flow Cytometry

A single-cell suspension was prepared by filtering the homogenate using a 40-μm pore cell strainer (BD Pharmingen, San Jose, CA, USA). Cells were incubated with Fc Block anti-CD16/32 (IC1918F, BD Pharmingen) and stained with fluorescein isothiocyanate-conjugated anti-fibronectin or isotype control (IC002F, R&D Systems) for 1 h. Fibronectin-positive cells and E-cadherin-positive cells were analyzed using a BD FACS Diva instrument (version 8.0; BD Biosciences). Fibrosis was induced in hTECs with 2 ng/mL rTGF-β (R&D Systems) for 48 h. To evaluate the role of CCL8 in fibrosis, rTGF-β-stimulated hTECs were simultaneously treated with an anti-CCL8 monoclonal antibody (50 or 100 ng/mL; Invitrogen). To quantify cell fibrosis and adhesion, cells were harvested and stained with antibodies against fibronectin (Invitrogen) and E-cadherin (R&D Systems) according to the manufacturers’ protocols. Apoptosis and necrosis were quantified by flow cytometry using an annexin V/propidium iodide assay. hTECs stained with propidium iodide and FITC-conjugated annexin V were incubated for 30 min in the dark, followed by analysis with the BD FACS Diva instrument.

### 2.8. Quantitative Real-Time PCR

Total RNA was isolated from hTECs and kidney tissues using TRIzol reagent (Thermo Fisher Scientific, Waltham, MA, USA). cDNA synthesis was performed using a reverse transcription kit (Promega, Madison, WI, USA) and a C1000 thermal cycler (Bio-Rad, Hercules, CA, USA). Subsequently, quantitative PCR was performed using a LightCycler-480 instrument II (Roche Molecular Systems Inc., Basel, Switzerland). Fibronectin, IL-8, CCR8, CCR2, and P53 mRNA levels were analyzed using the comparative Ct method (ΔΔCt) after normalization to GAPDH. PCR primers used for qRT-PCR are listed in Appendix A.

### 2.9. Confocal Microscopic Examination

rTGF-β-stimulated hTECs treated with or without anti-CCL8 monoclonal antibody (Invitrogen) were washed with Phosphate-buffered saline and fixed in 4% paraformaldehyde for 20 min. Following fixation, the cells were permeabilized with 0.3% Triton X and stained with antibodies against CCL8 (Biorbyt, St Louis, MO, USA) and CCR2 (Lifespan Biosciences) in a blocking agent overnight at 4 °C. Alexa 488/555-conjugated probes (Invitrogen) were used as secondary antibodies, and 4′,6-diamidino-2-phenylindole (DAPI; Invitrogen) was used to counterstain the nuclei. The primary antibodies were omitted in the negative controls. Immunofluorescence images were acquired with a confocal microscope (Leica TCS SP8, Leica Microsystem GmbH, Wetzlar, Germany).

### 2.10. Animals and Unilateral Ureteral Obstruction Model Establishment

Animal experiments were performed with the approval of the Institutional Animal Care and Use Committee (IACUC) of Seoul National University Hospital. C57BL/6 (WT) mice (The Jackson Laboratory, ME, USA) weighing 20 g were randomly divided into 4 groups (*n* = 6/group), which consisted of sham groups (1- and 2-week models) and unilateral ureteral obstruction (UUO) groups (3-day, and 1- and 2-week models). The UUO operation procedure has been described in detail in supplementary methods. After 3, 7, or 14 days after UUO, the mice were sacrificed, and the UUO left kidneys were collected for further analyses. Formalin-fixed, paraffin-embedded tissues were cut into 4-μm sections. For assessing fibrosis and apoptosis, staining was performed using Sirius Red and antibodies against Bax (Santa Cruz Biotechnology) and CCL8. 3–5 fields (×40) for each sample were randomly selected and quantified using computer-based morphometric analysis (Qwin 3). Scoring was performed in a blinded manner using the mean values of the positive areas (%).

### 2.11. Statistical Analysis

Chi-squared test was used to compare categorical data, such as frequencies and proportions. Continuous variables were expressed as mean ± standard deviation, or standard error of the mean where appropriate, and were compared using either the *t*-test or one-way analysis of variance test. Mann–Whitney U test or Kruskal–Wallis test was used to analyze nonnormally distributed variables, expressed as medians with an interquartile range. Pearson correlation coefficients were calculated to explore the linear relationship between serum CCL8 levels and various clinical parameters. All statistical analyses were performed using SPSS version 22 (IBM software, Armonk, NY, USA) and GraphPad Prism 9.0.2 (GraphPad Software, San Diego, CA, USA). Statistical significance was determined at *p* < 0.05.

## 3. Results

### 3.1. Correlation between Serum CCL8 ELISA Results and Clinico-Pathological Variables in CKD

The baseline characteristics of each of the three groups according to the diagnosis are shown in Table 1 (*n* = 157). BMI, serum C-reactive protein levels, and serum CCL 8 levels showed no significant differences among the three groups. However, compared to patients with IgAN, those with DN were older; had lower serum hemoglobin levels and eGFR; and higher blood urea nitrogen and serum creatinine levels and uPCR. Compared to patients in the IgAN group, those in the normal group were younger and showed lower blood urea nitrogen and serum creatinine levels and uPCR, and higher eGFR.

The relationship between clinical variables and serum CCL8 levels was investigated in IgAN and DN patients and healthy controls (Figure 1). The type of diagnosis, CKD stage according to eGFR, and uPCR were included in the analysis. In all patients, serum CCL8 levels were significantly higher in those at CKD stage 3–5 (*p* = 0.049; Figure 1b) and those with a uPCR ≥ 1 group (*p* = 0.021; Figure 1c). In IgAN patients, serum CCL8 levels were significantly higher in the uPCR ≥ 1 group (*p* = 0.032; Figure 1e).

Serum CCL8 levels were compared among IgAN and DN patients and healthy controls, to investigate their relationship with pathological factors identified in kidney biopsy results, including interstitial fibrosis (Figure 2a–c), interstitial atrophy (Figure 2d–f), and interstitial inflammation (Figure 2g–i). Serum CCL8 levels were significantly correlated with interstitial fibrosis, especially with marked-diffuse fibrosis, in all patients and patients with IgAN (*p* < 0.01; Figure 2a,c).

### 3.2. Localization of CCL8 and CCR2-Positive Cells in the Kidney by Immunohistochemical Analysis

The baseline characteristics of each of the three groups according to the diagnosis are shown in Table 2 (*n* = 83). Compared to patients in the IgAN group, those in the normal group were younger and had lower BMI, serum creatinine levels, and uPCR, and higher eGFR and serum albumin levels. CCL8 was found to be expressed in 16.4 ± 15.0% of cells in the kidney. Immunohistochemical analysis of kidney tissue sections showed that CCL8 was expressed predominantly in tubular cells and the inflamed tubulointerstitium. CCL8-positive cells were found in tubulointerstitial areas, but rarely in the glomeruli (Figure 3a,b).

CCL8 expression levels were significantly higher in patients with IgAN or DN than in healthy controls (*p* < 0.01; Figure 3c). CCL8 expression correlated with renal function at the time of biopsy, increasing significantly with advancing disease stage, especially in CKD stages 3 and 4 (*p* < 0.01; Figure 3d). The percentage of CCL8-positive areas correlated with uPCR (Figure 3e,f). CCL8 expression was significantly higher in patients with a marked degree of tubular atrophy, interstitial fibrosis, and interstitial inflammation (Figure 3g–i).

The correlation between serum CCL8 levels and the number of CCL8-positive cells at the same time point was assessed in 45 patients. Serum CCL8 levels positively correlated with CCL8 expression levels from immunohistochemical analysis (*p* = 0.039; Figure 3j).

We further investigated the tissue expression of CCR2, which is a known counterpart of CCL8. CCR2 immunohistochemical analysis in patients with IgAN in CKD showed that CCR2-positive cells were predominantly located in the periglomerular and inflamed tubulointerstitial area, especially in advanced CKD stages (Figure 4a). In immunofluorescence staining after induction of fibrosis with rTGF-β in hTECs, the increased expression regions of CCL8 and CCR2 were correlated and alleviation of expression was also observed in both after blocking CCL8 (Figure 4b). CCR2 expression levels were significantly higher in patients with CKD than in healthy controls, increasing significantly with advancing disease stages, especially in CKD stages 3 and 4 (*p <* 0.001; Figure 4c,d). The percentage of CCR2 expression was positively correlated with that of the CCL8 expression in kidney tissue (Figure 4e).

In addition, we further investigated the tissue expression of CCR8, which is a possible counterpart of CCL8. CCR8 expression levels were significantly higher in patients with CKD than in healthy controls, increasing significantly with advancing disease stages, especially in CKD stages 3 and 4 (*p* < 0.001; Appendix A). The percentage of CCR8-positive areas positively correlated with that of the CCL8-positive area (Appendix A).

### 3.3. TGF-β Induces Fibrosis in Primary Cultured hTECs

To confirm that CCL8 is expressed during kidney fibrosis in hTECs, the cells were treated with rTGF-β, and CCL8 protein expression was evaluated by western blotting. CCL8 was upregulated by rTGF-β treatment, which was accompanied by an increase in fibronectin and a decrease in E-cadherin and BCL-2 expression (Figure 5a,b). These results indicated that TGF-β induced fibrosis in hTECs via the upregulation of CCL8.

### 3.4. Effect of CCL8 Blockade and rCCL8 Induced by rTGF-β in hTECs

To evaluate the role of CCL8 in fibrosis, hTECs were simultaneously treated with anti-CCL8 mAb and rTGF-β. CCL8 blockade mitigated the rTGF-β-induced increase in fibronectin and CCR2 and decreased E-cadherin and BCL-2 protein levels (all *p* < 0.05) (Figure 5b). There was no significant difference in the expression of CD44. Based on these results, TGF-β induced kidney fibrosis in hTECs, an effect that was abrogated by blocking CCL8 function.

However, in the case of rCCL8, rCCL8 alone had no significant effect on inducing fibrosis. Simultaneous treatment with rTGF-β and rCCL8 enhanced the rTGF-β-induced increase in fibronectin expression (*p* < 0.05) (Figure 5c). The decrease in E-cadherin and BCL-2 expression induced by rTGF-β was not significantly changed by rCCL8 treatment, as determined by western blotting. Moreover, there were no significant differences in CD44 expression.

Fluorescence-activated cell sorting analysis of isolated primary cultured hTECs (Figure 5d), after induction of fibrosis with rTGF-β, there was a significant increase in fibronectin cell populations and a decrease in E-cadherin cell populations. Simultaneous treatment with anti-CCL8 mAb alleviated the rTGF-β-induced increase in fibronectin in a dose-dependent manner and decreased E-cadherin cell populations. The anti-apoptotic effect of the anti-CCL8 mAb was also demonstrated by Annexin V/propidium iodide staining using hTECs (Figure 5e). When treated with rTGF-β, the number of apoptotic cells increased significantly compared with the number of apoptotic cells after control IgG treatment.

Furthermore, CCR2/CCR8, positive correlation factor with CCL8; IL-8, a pro-inflammatory cytokine; fibronectin, a marker of fibrosis; and P53, an apoptosis marker, were analyzed by quantitative reverse transcription PCR. The mRNA expression levels of the indicated markers showed similar expression patterns to those observed by western blotting. The levels of all analyzed markers increased after treatment with rTGF-β, and these increases were significantly less pronounced, in a dose-dependent manner, after blocking CCL8 function (Figure 5f). These results indicated that the upregulation of CCL8 contributed to renal fibrosis and apoptosis.

### 3.5. Expression of CCL8 and Fibrosis- and Apoptosis-Related Markers in a UUO Model

The immunohistochemical analysis showed that the expression of type I collagen, Bax, a marker of apoptosis, and CCL8 was strongly induced 1 week after the UUO and became more pronounced 2 weeks after the UUO. In the morphometric analysis, the three indicated marker-positive area was increased significantly over time (Figure 6b), which was in agreement with our in vitro findings.

## 4. Discussion

Our results are the first to suggest a role for CCL8 in the kidney fibrosis process. Higher serum CCL8 levels were associated with advanced CKD stage, urine protein creatinine ratio, and kidney fibrosis in biopsy samples, and CCL8 was highly expressed in kidney tissue. Consistent with these in vitro findings, CCL8 was highly expressed in kidney tissue in a mouse model of CKD. Inhibition of CCL8 signaling using an anti-CCL8 mAb abrogated rTGF-β-induced kidney fibrosis in primary cultured hTECs.

CCL8 is a CC chemokine that belongs to a family of small cytokines. The CCL8 protein starts out as a 109-amino-acid precursor, which is then cleaved to create mature CCL8 with 75 amino acids [19,23]. CCL8 chemotactically attracts and activates a variety of immune cells, including mast cells, eosinophils, basophils, monocytes, T cells, and NK cells, all of which are implicated in the inflammatory response [24,25]. Fibrosis of diverse organs and tissues is regarded as a pathological consequence of a chronically altered wound healing response by the recruitment of immune cells. It is known that CCL8 is produced via toll-like receptors (TLRs) when fibroblasts and endothelial cells that are co-stimulated with IL1-β and interferon (IFN)-γ [26]. Lee et al. reported that CCL8 was overexpressed in IPF lungs compared to normal lungs, and CCL8 could be a candidate biomarker for the diagnosis and prediction of survival in IPF [20]. Although several chemokines and chemokine receptors, such as CXCL, CCL, CX3CL, and XCL groups, have been shown to be associated with kidney fibrosis [27], the role of CCL8 has not been previously investigated.

The most important finding of our study is the validation of the anti-fibrotic and anti-apoptotic effects of CCL8 blockade in the kidney fibrosis process. Inflammation is a key factor in the pathogenesis of CKD, and, at the molecular level, TGF-β is known to be a key factor in the kidney fibrosis pathway, including inflammatory reactions [28,29,30]. It was previously thought that resident renal fibroblasts and infiltrating inflammatory cells are crucial mediators of kidney fibrosis. However, recent studies have shown that EMT of tubular epithelial cells is critical for the induction of fibrosis [31,32]. Following repeated exposure to certain growth factors, such as TGF-β, tubular epithelial cells transform into fibroblasts via EMT. In this process, the breakdown of intercellular connections begins with the downregulation of adhesion molecules such as E-cadherin [33,34]. Previous studies have shown that apoptosis plays a crucial role in the development of TGF-β-induced injury in kidney tubular epithelial cells [35,36], and it has been reported that miRNAs play a role in some physiological processes, including cell differentiation, proliferation, apoptosis, stress resistance, and angiogenesis [37]. It has also been shown that upregulation of CCL8 leads to apoptosis in association with miRNAs [38,39]. In the present study, treatment of cultured hTECs with rTGF-β induced EMT and reduced the level of E-cadherin, which was accompanied by enhanced fibronectin expression. CCL8 blockade markedly ameliorated fibrosis and apoptosis in rTGF-β-induced hTECs. Thus, inhibiting EMT and anti-apoptotic effects by blocking CCL8 can prevent kidney fibrosis induced by TGF-β.

CCR8 has been recently identified as a receptor for CCL18 [40,41,42]. This chemokine receptor is expressed on T helper type 2 cells and regulatory T cells as well as monocytes and macrophages in kidney [43]. In the present study, immunohistochemistry results showed co-expression and positive correlation of CCL8-CCR2 axis and CCR8 expression in the kidney of patients with CKD. Our results suggest that CCL8-CCR2 axis might activate CCR8 or induce their inflammatory reactions of CCR8 in kidney fibrosis.

There are several limitations to our study that should be mentioned. First, we did not investigate the effects of a CCL8 inhibitor in vivo. However, the TGF-β-induced fibrosis in vitro model clearly demonstrated the effects of CCL8 inhibition on kidney fibrosis. Second, we did not examine the differences in change over time in the in vitro model, and it was impossible to confirm the long-term association with CCL8 levels. Third, we did not investigate the detailed mechanisms underlying the association between CCL8 and fibrosis and the apoptosis pathway or the CCL8-CCR2 axis and CCR8 interaction during the progression of CKD. Therefore, further studies investigating these mechanisms should be performed to validate our findings fully.

## 5. Conclusions

Taken together, our results demonstrate that the CCL8 pathway is associated with an increased risk of kidney fibrosis, and that CCL8 blockade can ameliorate kidney fibrosis and apoptosis. Thus, therapeutic strategies that inhibit CCL8 function may be effective in preventing the development and progression of CKD.

## Figures and Tables

**Figure 1 cells-11-00658-f001:**
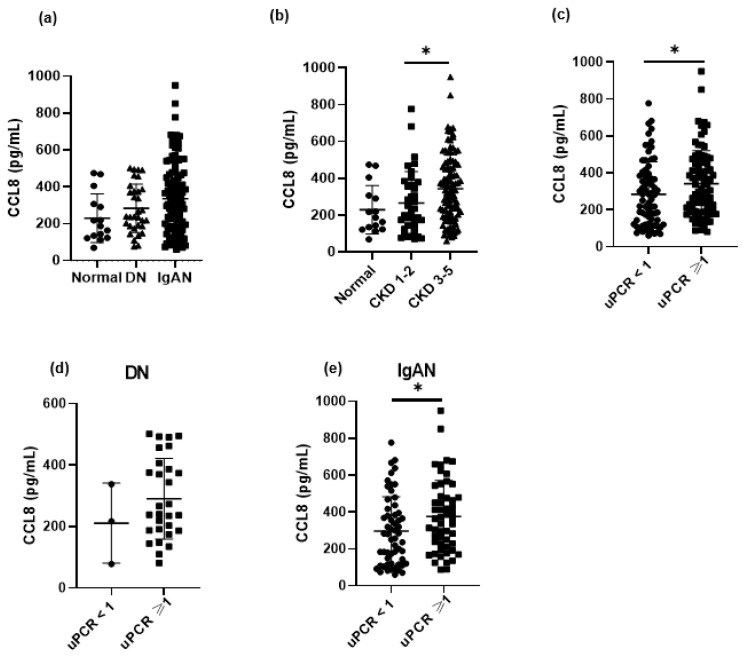
Serum CCL8 level were compared among patients with IgA nephropathy (IgAN, *n* = 110), diabetic nephropathy (DN, *n* = 32), and healthy controls (HC, *n* = 15) to investigate relation with clinical variables, such as type of diagnosis, chronic kidney disease stage and urine protein creatinine ratio (uPCR). In total patients, serum CCL8 level showed no difference according to diagnosis, but significantly higher in chronic kidney disease stage 3–5 group and uPCR ≥ 1 group (**a**–**c**). No difference in serum CCL8 level was observed according to uPCR level in DN patients, but significantly higher in IgAN patients with uPCR ≥ 1 (**d**,**e**). * *p* < 0.05.

**Figure 2 cells-11-00658-f002:**
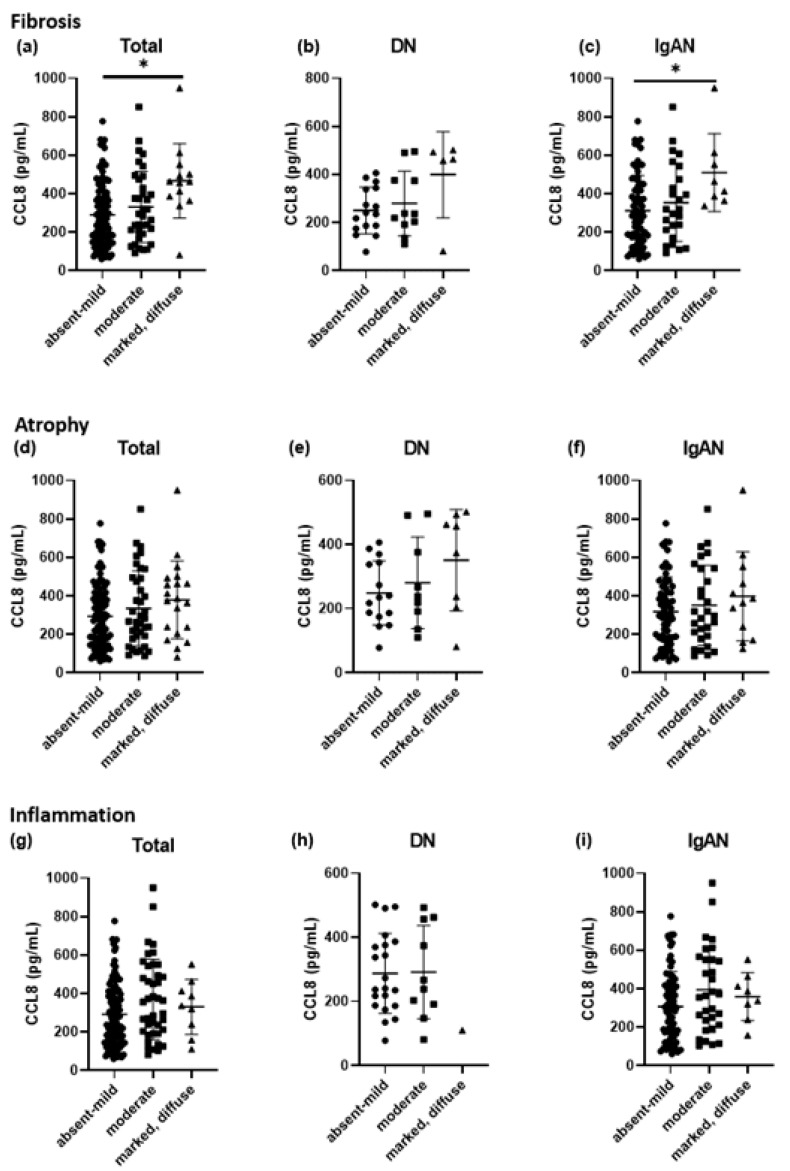
Serum CCL8 level were compared among patients with IgA nephropathy (IgAN, *n* = 110), diabetic nephropathy (DN, *n* = 32), and healthy controls (HC, *n* = 15) to investigate relation with pathologic factor of kidney biopsy result. Interstitial fibrosis (**a**–**c**), tubular atrophy (**d**–**f**), and interstitial inflammation (**g**–**i**) were investigated. In total patients and patients with IgAN, serum CCL8 level was significantly correlated with interstitial fibrosis (**a**,**c**). * *p* < 0.05.

**Figure 3 cells-11-00658-f003:**
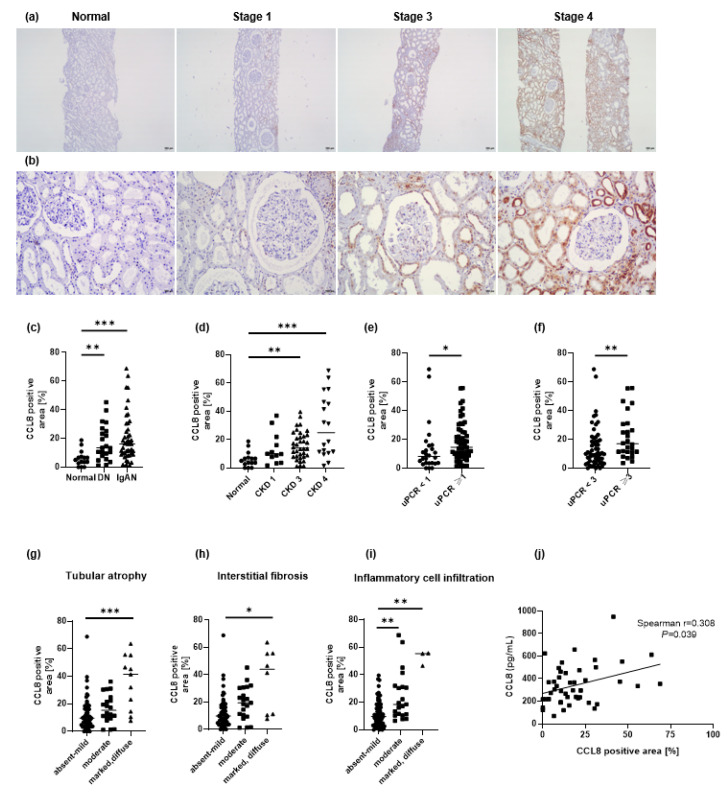
Representative photographs of kidney sections of healthy controls (HC), chronic kidney disease stage 1, 3, and 4 of IgA nephropathy (IgAN) patients (original magnifications, Panel (**a**), ×40 and (**b**), ×200). CCL8 immunohistochemical analysis in patients with IgAN showed that CCL8-positive cells were predominantly located in the periglomerular and inflamed tubulointerstitium (**a**,**b**). Quantification of CCL8-positive cells was performed by computer-based morphometric analysis. CCL8 expression were compared among patients with IgAN (*n* = 46), diabetic nephropathy (DN, *n* = 22), and HC (*n* = 15) (**c**), which correlated with kidney function at the time of biopsy. The expression of CCL8 increased significantly with advancing stages, especially in chronic kidney disease stage 3 and 4 (**d**). The level of CCL8 positive area was correlated with urine protein/creatinine ratio (uPCR) (**e**,**f**). CCL8 expressions were significantly higher in marked degree of tubular atrophy (**g**), interstitial fibrosis (**h**) and interstitial inflammation (**i**). In 45 patients who were examined for serum CCL8 and number of CCL8-positive cell simultaneously, serum CCL8 levels positively correlated with CCL8 expression levels from immunohistochemical analysis (**j**). * *p* < 0.05, ** *p* < 0.01, and *** *p* < 0.001.

**Figure 4 cells-11-00658-f004:**
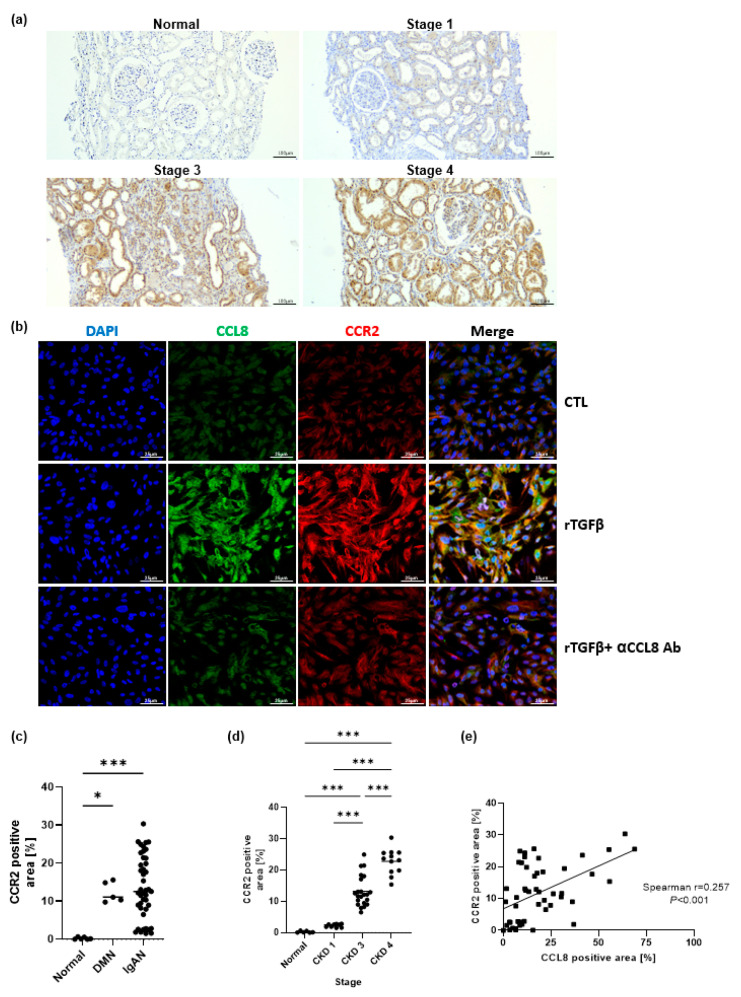
Representative photographs of kidney sections of healthy controls (HC), chronic kidney disease stage (CKD) 1, 3, and 4 of IgA nephropathy (IgAN) patients (original magnifications, Panel (**a**), ×100). CCR2 immunohistochemical analysis in patients with IgAN showed that CCR2-positive cells were predominantly located in the periglomerular and inflamed tubulointerstitium (**a**). After induction of fibrosis with rTGF-β in hTECs, the increased expression regions of CCL8 and CCR2 were correlated in immunofluorescence staining and alleviation of expression was also observed in both after blocking CCL8 (original magnifications, Panels b, ×400 (**b**)). CCR2 expression levels were significantly higher in patients with IgAN and advanced CKD stages (**c**,**d**). The percentage of CCR2 positively correlated with that of the CCL8 (**e**). * *p* < 0.05, and *** *p* < 0.001.

**Figure 5 cells-11-00658-f005:**
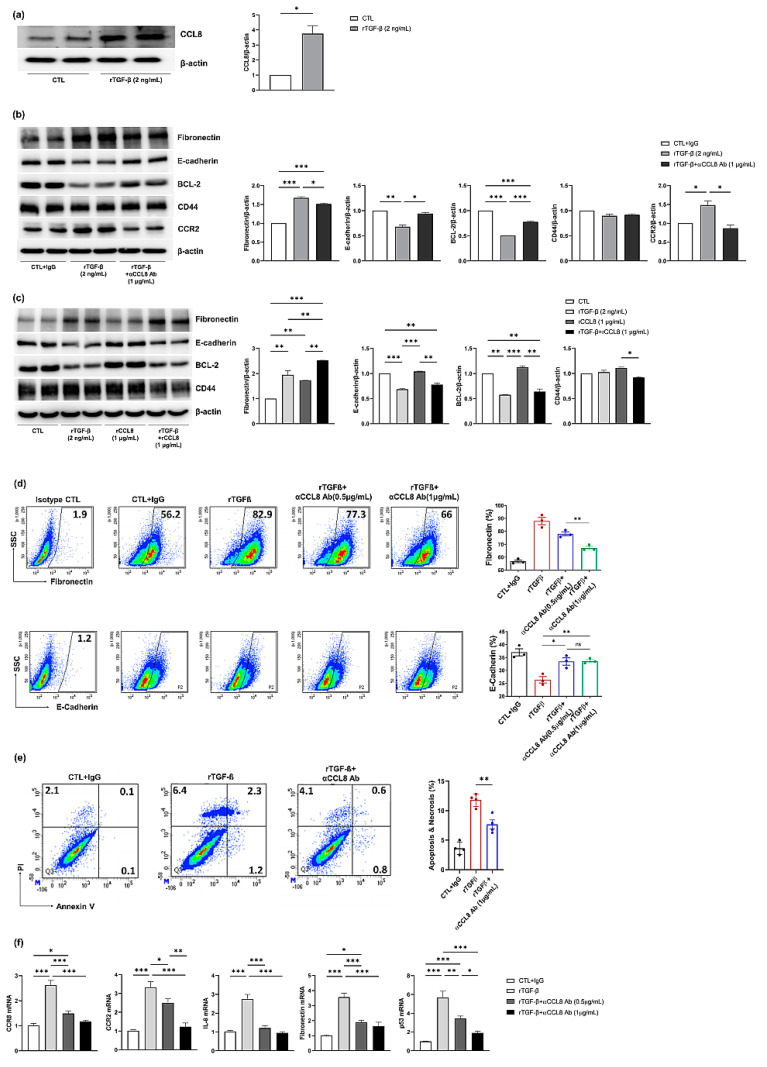
Western blot analysis of primary cultured hTECs after induction of fibrosis with rTGF-β (2 ng/mL). CCL8 was up-regulated by rTGF-β treatment (**a**). rTGF-β increased fibronectin and CCR2 and decreased E-cadherin and BCL-2 protein expression. The simultaneous treatment with anti-CCL8 mAb at 1.0 μg/mL, mitigated the rTGF-β-induced increase in fibronectin and CCR2 and decrease E-cadherin and BCL-2 protein levels. Bars represent means ± SDs (**b**). rCCL8 alone had no direct effect on increasing in fibronectin expression. The simultaneous treatment with rTGF-β and rCCL8 (1 μg/mL), enhanced the rTGF-β-induced increase in fibronectin and decrease E-cadherin and BCL-2 protein expression (**c**). Representative FACS analysis of isolated primary cultured hTECs. After induction of fibrosis with rTGF-β (2 ng/mL) for 48 h, there were significant increase in fibronectin cell populations and decrease in E-cadherin cell populations. The simultaneous treatment with anti-CCL8 mAb (50 and 100 ng/mL) alleviated the rTGF-β-induced increase in fibronectin in a dose-dependent manner and decrease E-cadherin cell populations (**d**). The antiapoptotic effect of the anti-CCL8 mAb was also demonstrated by Annexin V/propidium iodide staining assay (**e**). Quantitative real-time PCR analysis of CCR8, CCR2, IL-8, fibronectin, and P53 with anti-CCL8 mAb (50 and 100 ng/mL) treatment (**f**). * *p* < 0.05, ** *p* < 0.01, and *** *p* < 0.001.

**Figure 6 cells-11-00658-f006:**
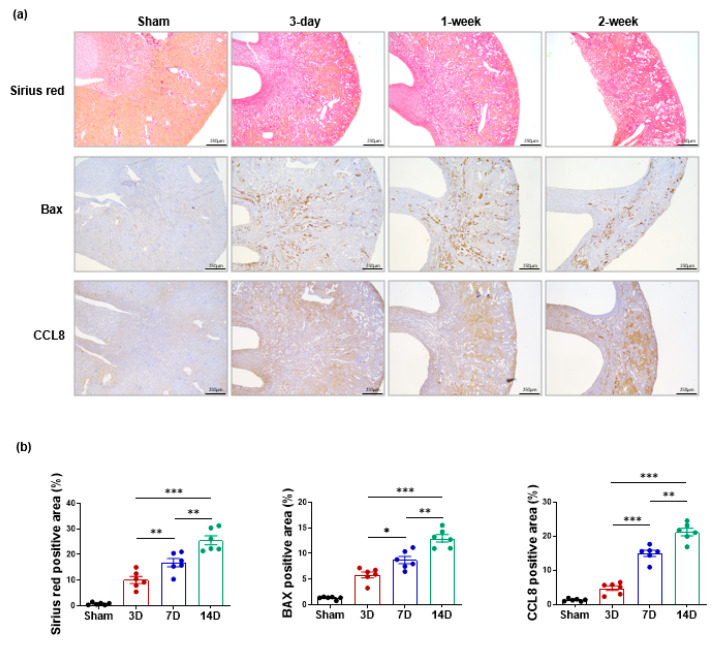
The immunohistochemical analysis showed that the expression of type I collagen, Bax, a marker of apoptosis, and CCL8 was strongly induced 1 week after the unilateral ureteral obstruction (UUO) and became more pronounced 2 weeks after the UUO (original magnifications, Panel (**a**), ×40). Quantification was performed by computer-based morphometric analysis. The three indicated marker-positive area was increased significantly over time (**b**). * *p* < 0.05, ** *p* < 0.01, and *** *p* < 0.001.

**Table 1 cells-11-00658-t001:** Baseline characteristics of serum CCL8 ELISA group.

	IgAN (*n* = 110)	DN (*n* = 32)	HC (*n* = 15)	*p*			
				Group	IgAN vs. HC	IgAN vs. DN	DN vs. HC
Age (years)	45.95 ± 15.77	54.78 ± 15.04	38.93 ± 16.47	0.003	0.136	0.005	0.004
Male (%)	64 (58.2%)	20 (62.5%)	7 (46.7%)	0.589			
Body mass index (kg/m^2^)	24.03 ± 3.87	24.47 ± 3.43	23.31 ± 4.36	0.631	0.559	0.595	0.244
CKD stage 1–2	28 (25.5%)	9 (28.1%)					
CKD stage 3	55 (50.0%)	15 (46.9%)					
CKD stage 4–5	27 (24.5%)	8 (25.0%)					
Hemoglobin (g/dL)	12.39 ± 1.99	10.89 ± 1.68	13.42 ± 1.48	<0.001	0.066	<0.001	<0.001
CRP (mg/dL)	0.28 ± 0.82	0.10 ± 0.13	0.21 ± 0.47	0.634	0.704	0.432	0.375
Albumin (g/dL)	3.84 ± 0.57	3.61 ± 0.57	4.21 ± 0.30	<0.001	0.004	0.016	<0.001
BUN (mg/dL)	23.35 ± 13.21	28.81 ± 14.18	12.00 ± 4.16	<0.001	<0.001	0.018	<0.001
Creatinine (mg/dL)	1.80 ± 1.52	2.00 ± 1.42	0.77 ± 0.17	<0.001	<0.001	0.513	<0.001
eGFR (CKD-EPI, mL/min/1.73 m^2^)	53.01 ± 32.26	47.59 ± 27.43	104.27 ± 18.53	<0.001	<0.001	0.529	<0.001
uPCR	1.59 ± 1.97	3.21 ± 2.02	0.65 ± 0.73	<0.001	<0.001	<0.001	<0.001
Serum CCL8 ELISA (pg/mL)	335.27 ± 194.08	283.46 ± 131.25	229.55 ± 131.59	0.103	0.05	0.335	0.138

Data are presented as mean ± SD or *n* (%). Comparison was performed using Mann–Whitney U test or Kruskal–Wallis test. Abbreviations: CCL8, chemokine (C-C motif) ligand 8; CKD, chronic kidney disease; CRP, C-reactive protein; DN, diabetic nephropathy; eGFR, estimated glomerular filtration rate; HC, healthy controls; IgAN, IgA nephropathy; uPCR, urine protein/creatinine ratio.

**Table 2 cells-11-00658-t002:** Baseline characteristics of tissue CCL8 expression immunohistochemistry group.

	IgAN (*n* = 46)	DN (*n* = 22)	HC (*n* = 15)	*p*			
				Group	IgAN vs. HC	IgAN vs. DN	DN vs. HC
Age (years)	47.6 ± 14.9	54.9 ± 12.4	37.3 ± 16.7	0.003	0.061	0.180	0.002
Male (%)	21 (45.7%)	15 (68.2%)	6 (40.0%)	0.146			
Body mass index (kg/m^2^)	24.7 ± 4.4	24.5 ± 3.9	21.2 ± 3.5	0.023	0.023	1.000	0.070
CKD stage 1	10 (21.7%)	4 (18.2%)					
CKD stage 3	24 (52.2%)	12 (54.6%)					
CKD stage 4	12 (26.1%)	6 (27.3%)					
Hemoglobin (g/dL)	12.3 ± 1.9	12.0 ± 2.1	13.7 ± 1.9	0.046	0.083	1.000	0.056
Albumin (g/dL)	3.6 ± 0.6	3.8 ± 0.4	4.1 ± 0.4	0.009	0.008	0.664	0.203
Creatinine (mg/dL)	1.67 ± 0.85	1.52 ± 0.52	0.84 ± 0.32	0.001	0.001	1.000	0.018
eGFR(CKD-EPI, mL/min/1.73 m^2^)	55.8 ± 35.7	44.5 ± 22.9	95.7 ± 28.6	<0.001	<0.001	0.517	<0.001
uPCR	2.9 ± 2.5	3.1 ± 2.5	0.6 ± 0.4	0.002	0.003	1.000	0.006
CCL8 positive area (%)	20.2 ± 16.7	16.8 ± 12.4	6.0 ± 5.4	0.005	0.004	1.000	0.077

Data are presented as mean ± SD or *n* (%). Comparison was performed using Mann–Whitney U test or Kruskal–Wallis test. Abbreviations: CCL8, chemokine (C-C motif) ligand 8; CKD, chronic kidney disease; DN, diabetic nephropathy; eGFR, estimated glomerular filtration rate; HC, healthy controls; IgAN, IgA nephropathy; uPCR, urine protein/creatinine ratio.

## Data Availability

The data presented in this study are available in the main article.

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
