# Peer review of "Chemokine (C-C Motif) Ligand 8 and Tubulo-Interstitial Injury in Chronic Kidney Disease"

_cells, 2022, doi:10.3390/cells11040658_

Round 1

Reviewer 1 Report

In this manuscript, Jangwook Lee and colleagues reported their findings indicating the novel role of chemokine (C-C motif) ligand 8 (CCL8) in kidney fibrosis and chronic kidney disease (CKD). In the beginning, CCL8 was identified as the most upregulated chemokine in IgA nephropathy (IgAN) compared to healthy controls (data not shown). To investigate the potential role of CCL8, the authors used clinical samples, an in vitro cell model and a mouse model of UUO. The results showed that serum CCL8 levels were associated with advanced CKD stage and kidney fibrosis in biopsy samples. Consistent with these in vitro findings, CCL8 was highly expressed in kidney tissue in a mouse model of CKD. Inhibition of CCL8 signaling using an anti‐CCL8 mAb abrogated rTGF-β-induced kidney fibrosis in primary cultured hTECs, indicating a novel role of CCL8 in the kidney fibrosis process.

This well-written paper is of enough novelty and workload. However, there are some minor and major aspects of this manuscript that are not convincing and must be improved.

Major:

  1. Page 2, line 53-57

To identify new markers of kidney fibrosis, the authors performed a prospective study in patients with IgAN by analyzing microarrays of kidney biopsies from paraffin-embedded tissues and identified CCL8 as the most upregulated chemokine. However, NO statistical measures or results were provided to accompany this statement. 

  1. Page 2, line 67-70; Page 12, line 386-391; Fig 3 & 4

In Page 2 line 67-70, the authors mentioned that CCL8 serves as an agonist for CCR8. However, in reference 21 Islam et al. reported that mouse CCL8, NOT human CCL8, acted as an agonist for CCR8. In this paper (Islam et al. Nat Immunol 2011), the authors explicitly mentioned that “human CCL8, a known CCR2 agonist, does not activate CCR8”, “Ccr8 deficiency had no effect on Th2-R2A cell migration to human CCL8”, “human CCL8 did not induce calcium flux or migration of human CCR8–transfected cells, …, confirming that human CCL8 was not a human CCR8 agonist”. As a matter of fact, “human CCL8/MCP2 is a different cytokine” (See PMID:34307414 and 33182504), and activates different receptors compared to mouse CCL8.

In this paper, the authors assumed that CCR8 was a possible counterpart of CCL8, and then investigated the correlation between the expression of human CCL8 and CCR8 in clinical samples. In my humble opinion, this might be possible, but the assumption itself is not sufficiently rigorous. 

Although the results about CCL8 and CCR8 seem not surprising, this paper is the first to demonstrate the relationship between human CCL8 and CCR8. To confirm this hypothesis (or conclusion on page 12 line 390 “Our results suggest that CCL8 might act through CCR8 or induce their inflammatory reactions by CCR8 in kidney fibrosis.”), further experiments should be added and this may be inevitable and necessary.

Minor:

  1. Table 1 & Table 2

In the serum CCL8 ELISA group, the patients were divided into 3 groups according to CKD stage, which were stage1-2, 3 and 4-5, while in the tissue CCL8 expression immunohistochemistry group there were 3 different CKD stage groups including stage 1, 3 and 4. Why is it different?

  1. Figure 3, figure legend

Figure legend for fig. 3m is missing.

Author Response

Dear reviewer,

First of all, we appreciate the very kind consideration of the editorial board and apology for the late response due to additional experiments regarding the issue raised by reviewers.

We think that our manuscript has been improved greatly through your comments. We have made several changes to the manuscript accordingly. Your comments are in italics, our responses are in blue, and changes to the manuscript are in red.

We hope that our revised manuscript is suitable for publication in Cells.

Sincerely,

Reviewer 2 Report

The Authors experimentally evaluated serum levels and tissue expression of CCL8 in CKD patients, to analyze CCL8 association with kidney fibrosis.

A fundamental piece of information missing is inclusion and exclusion criteria of patients who underwent kidney biopsy and provided the samples.

It may be interesting to compare the findings of this study with other experiences regarding kidney disease and macrophage-related chemokines  (a recent and useful review was performed by Cantero-Navarro and coll, doi: 10.3389/fmed.2021.6880609) or the role of macrophages as biomarker in conditions such as resistant hypertension (Gembillo and coll, doi: 10.1007/s11255-021-02904-9).

Author Response

(The authors gave the same response as above.)
